# DNA-Vaccine-Induced Immune Response Correlates with Lower Viral SARS-CoV-2 Titers in a Ferret Model

**DOI:** 10.3390/vaccines10081178

**Published:** 2022-07-25

**Authors:** Mirco Compagnone, Eleonora Pinto, Erika Salvatori, Lucia Lione, Antonella Conforti, Silvia Marchese, Micol Ravà, Kathryn Ryan, Yper Hall, Emma Rayner, Francisco J. Salguero, Jemma Paterson, Matteo Iannacone, Raffaele De Francesco, Luigi Aurisicchio, Fabio Palombo

**Affiliations:** 1Neomatrix, 00128 Rome, Italy; compagnone@takisbiotech.it; 2Takis, 00128 Rome, Italy; pinto@takisbiotech.it (E.P.); salvatori@takisbiotech.it (E.S.); lione@takisbiotech.it (L.L.); 3Evvivax, 00128 Rome, Italy; 4INGM-Istituto Nazionale di Genetica Molecolare “Romeo ed Enrica Invernizzi”, 20122 Milan, Italy; marchese@ingm.org (S.M.); defrancesco@ingm.org (R.D.F.); 5Division of Immunology, Transplantation and Infectious Diseases, IRCCS San Raffaele Scientific Institute, 20132 Milan, Italy; rava.micol@hsr.it (M.R.); iannacone.matteo@hsr.it (M.I.); 6UK Health Security Agency (UKHSA), Porton Down, Salisbury SP4 0JG, UK; kathryn.ryan@ukhsa.gov.uk (K.R.); yper.hall@ukhsa.gov.uk (Y.H.); emma.rayner@ukhsa.gov.uk (E.R.); javier.salguero@ukhsa.gov.uk (F.J.S.); jemma.paterson@ukhsa.gov.uk (J.P.); 7Vita-Salute San Raffaele University, 20132 Milan, Italy; 8Experimental Imaging Centre, IRCCS San Raffaele Scientific Institute, 20132 Milan, Italy; 9Department of Pharmacological and Biomolecular Sciences (DiSFeB), University of Milan, 20133 Milan, Italy

**Keywords:** SARS-CoV-2, ferrets, DNA vaccines, immune responses

## Abstract

The COVID-19 pandemic is entering a new era with the approval of many SARS-CoV-2 vaccines. In spite of the restoration of an almost normal way of life thanks to the immune protection elicited by these innovative vaccines, we are still facing high viral circulation, with a significant number of deaths. To further explore alternative vaccination platforms, we developed COVID-*e*Vax—a genetic vaccine based on plasmid DNA encoding the RBD domain of the SARS-CoV-2 spike protein. Here, we describe the correlation between immune responses and the evolution of viral infection in ferrets infected with the live virus. We demonstrate COVID-*e*Vax immunogenicity as means of antibody response and, above all, a significant T-cell response, thus proving the critical role of T-cell immunity, in addition to the neutralizing antibody activity, in controlling viral spread.

## 1. Introduction

SARS-CoV-2, the coronavirus responsible for severe acute respiratory syndrome, has been the objective of unprecedented vaccine development since 2020. In total, the WHO estimates that 14.9 million deaths have been directly or indirectly associated with SARS-CoV-2 infection. SARS-CoV-2 is spread through respiratory droplets and/or direct contact, and infects host cells via the angiotensin-converting enzyme 2 (ACE2) [1,2]. There are a variety of animal species vulnerable to SARS-CoV-2 infection, including macaques, cats, ferrets, hamsters, and transgenic mice expressing human ACE2 [3,4]. However, many of these models only partially mimic human COVID-19 symptoms, and illness in animals is typically mild, with varying levels of severity depending on the animal model. In addition to non-human primates, one of the most reliable COVID-19 models to date is the ferret, which has long been used in respiratory virus research [5,6].

We have recently shown the efficacy of COVID-*e*Vax—a genetic vaccine expressing the RBD domain of the SARS-CoV-2 spike protein [7]. The RBD sequence was modified and cloned in a DNA plasmid vector and delivered by electroporation (EP), resulting in strong immunological responses and protection against viral challenges in the K18-hACE2 mouse model. COVID-*e*Vax utilizes a delivery platform employed in experimental models, and has been shown to induce a potent antibody response in non-human primates [8], as well as T-cell immune responses in preclinical cancer models [9,10,11]. The immunogenic mechanism of the naked DNA plasmid delivered by electroporation (EP) is likely mediated by the enhancement of gene expression, along with the induction of danger signals by cytoplasmic DNA and the transient tissue damage caused by the electrical field utilized to allow the DNA to enter muscle cells. Similar results have been obtained by other research groups under similar electrical conditions. The advantage of naked DNA delivered by EP is that it enables the vaccine to be administered many times without limiting immune responses, which may potentially occur with a viral vector. A plasmid DNA vaccine (ZyCoV-D) against SARS-CoV-2 developed by Zydus Cadila has been approved for emergency use authorization (EUA) by the office of the Drug Controller General of India (DCGI), with a vaccination protocol consisting of three doses every 28 days [12].

Given the flexibility of plasmid DNA vaccines, which can be rapidly generated and adapted to emerging variants of concern (VOCs), we investigated the mechanisms of protection in ferrets as a model of mild human disease for SARS-CoV-2 [13]. Although COVID-*e*Vax resulted in a limited neutralization capacity, it primed the immune system against VOCs, and induced a strong T-cell response that was correlated with the reduction in viral RNA.

## 2. Material and Methods

### 2.1. Vaccination and Viral Challenge

The 7-month-old female ferrets (*Mustela putorius furo*) were obtained from a UK Home-Office-accredited supplier (Highgate Farm, High Peak, SK22 2JW UK). All experimental work was conducted under the authority of a UK Home-Office-approved project license that had been subject to local ethical review at UKHSA Porton Down by the Animal Welfare and Ethical Review Body (AWERB), as required by the Home Office Animals (Scientific Procedures) Act 1986. All animals in groups 1 and 2 were vaccinated with 400 μg of COVID-*e*Vax via the intramuscular route into the left quadriceps. This was immediately followed by EP delivered using a Cliniporator device (Igea, Italy) using N-10-4B needles. Animals in group 3 were not vaccinated. Days of vaccination were as depicted in Figure 1. Animals were approved for release into the study by the NACWO (Named Animal Care and Welfare Officer).

Animals were challenged with live SARS-CoV-2 virus, VERO/hSLAM cell passage 3 (Victoria/1/2020), at a target dose of 5.0 × 10^6^ plaque-forming units (PFUs), by the distribution of 0.5 mL of inoculum per nostril. An aliquot (0.2 mL) of the challenge virus was retained for confirmation of the challenge dose by plaque assay.

### 2.2. Clinical Data

Clinical signs were measured once daily at a consistent time of day during the vaccination phase of the study. Temperatures were recorded at the two extremes of the working day. Animals were weighed once daily. Post-challenge (pc) clinical observations were recorded twice daily, and the animals were weighed every day. Temperature monitoring (using a temperature chip) twice a day was continued until the study’s termination.

### 2.3. Molecular Tests

RNA was isolated from nasal washes and throat swabs. Samples were inactivated in AVL (Qiagen, Hilden, Germany) and ethanol. Downstream extraction was then performed using the BioSprint™ 96 One-For-All vet kit (Indical) and Kingfisher Flex platform, as per the manufacturer’s instructions. Tissue homogenate was centrifuged through a QIAshredder homogenizer (Qiagen) and supplemented with ethanol as per the manufacturer’s instructions. Downstream extraction from tissue samples was then performed using the BioSprint™ 96 One-For-All vet kit (Indical) and Kingfisher Flex platform, as per the manufacturer’s instructions. Reverse-transcription quantitative polymerase chain reaction (RT-qPCR) targeting a region of the SARS-CoV-2 nucleocapsid (N) gene was used to determine viral loads, and was performed using TaqPath™ 1-Step RT-qPCR Master Mix, CG (Applied Biosystems™), the 2019-nCoV CDC RUO Kit (Integrated DNA Technologies, Coralville, IA, USA), and a QuantStudio™ 7 Flex Real-Time PCR System. Undetected samples were assigned the value of <2.3 copies/μL, equivalent to the assay’s lower limit of detection (LLOD).

### 2.4. ELISA Assay

ELISA assay was performed on sera from vaccinated or untreated ferrets for antibody titration, as previously reported [7]. Briefly, the plates were functionalized by coating them with the RBD-6xHis protein at a concentration of 1 μg/mL and incubated for 18 h at 4 °C, and then blocked with 3% BSA–0.05% Tween 20-PBS for 1 h at room temperature. Ferret sera were then added at a dilution of 1/300 in 1% BSA–0.05% Tween 20-PBS and diluted 1:3 up to 1/218,700, in duplicate, and the plates were incubated for 18 h at 4 °C. After washing three times with 0.05% Tween 20-PBS, the secondary anti-ferret IgG conjugated with horseradish peroxidase (HRP) was added at a dilution of 1:40,000 in 1% BSA–0.05% Tween 20-PBS, and the plates were incubated for 1 h at room temperature. After washing three times with 0.05% Tween 20-PBS, the binding of the secondary antibody was detected by adding TMB (3,3′,5,5′ tetramethylbenzidine) liquid substrate (Merck, Italy). After incubation for 10 min at room temperature in the dark, Stop Reagent (Merck, Italy) was added, and the absorbance was measured at 450 nm using an ELISA reader. IgG antibody titers against the RBD protein were evaluated at two time points (day 14 and day 0). Endpoint titers were calculated by plotting the log10 OD against the log10 sample dilution. A regression analysis of the linear part of the curve allowed calculation of the endpoint titer. An OD of 0.2 was used as a threshold.

### 2.5. ELISpot Assay

For the detection of RBD-specific T-cell response, the Ferret IFN-γ ELISpot Kit (ALP) from Mabtech was used on frozen PBMCs collected from vaccinated and control ferrets at days 14 and 0. Then, the PBMCs were thawed in RPMI 1640 medium (Gibco, Waltham, MA, USA) supplemented with CTL medium (ImmunoSpot, Shaker Heights, OH, USA) and Benzonase (50 U/mL). After washing, the cells were resuspended in RPMI 1640 supplemented with penicillin/streptomycin (100 U/mL) and 20% fetal bovine serum, and then incubated for 2 h at 37 °C. Cells were then resuspended in RPMI 1640 supplemented with penicillin/streptomycin (100 U/mL) and 10% fetal bovine serum, and viable cells were plated at 2 × 10^5^ and 5 × 10^5^ cells/well. Cells were stimulated with a pool of RBD peptides (JPT) at a final concentration of 1 μg/mL for 16–18 h at 37 °C. Dimethyl sulfoxide (DMSO) and concanavalin A (ConA) were used as negative and positive stimulation controls, respectively. After overnight stimulation at 37 °C, plates were washed and incubated with biotinylated anti-mouse IFN-γ antibody (1:1000), and then washed and incubated for 2 h at room temperature with streptavidin-AP-conjugated antibody. After extensive washing, 50 μL/well of the substrate (NBT/BCIP-1 step solution, Pierce) was added to measure spot development. The plates were thoroughly washed with distilled water to stop the reaction. Plates were allowed to air-dry completely, and spots were counted using an automated ELISpot reader (Aelvis ELISpot reader, A.EL.VIS GmbH, Bremen, Germany).

### 2.6. Neutralization Assay

The HEK293TN-hACE2 cell line was generated by lentiviral transduction of HEK293TN cells (System Bioscience, Palo Alto, CA, USA). Lentiviral vectors were produced following a standard procedure based on calcium phosphate co-transfection with 3rd-generation helper and transfer plasmids. The following helper vectors (gifts from Prof. Didier Trono) were used: pMD2.G/VSV-G (Addgene #12259), pRSV-Rev (Addgene #12253), and pMDLg/pRRE (Addgene #12251). The transfer vector pLENTI_hACE2_HygR was obtained by cloning hACE2 from pcDNA3.1-hACE2 (a gift from Prof. Fang Li, Addgene #145033) into pLenti-CMV-GFP-Hygro (a gift from Prof. Eric Campeau and Paul Kaufman, Addgene #17446); hACE2 cDNA was amplified by PCR and inserted under the CMV promoter of the pLenti-CMV-GFP-Hygro after GFP excision with XbaI and SalI digestion; pLENTI_hACE2_HygR is now made available through Addgene (Addgene #155296).

To generate lentiviral particles pseudotyped with the SARS-CoV-2 spike protein, we constructed a series of expression plasmids each encoding a different SARS-CoV-2 spike protein variant. Briefly, for each variant, the corresponding C-terminal-deleted (19 amino acids) spike protein cDNA was cloned in pcDNA3.1. pLenti CMV-GFP-TAV2A-LUC Hygro was generated from pLenti CMV GFP Hygro (Addgene #17446) by addition of T2A-Luciferase via PCR cloning. To produce the pseudotyped lentiviral particles, 5 × 10^6^ HEK-293TN cells were plated in a 15 cm dish in complete DMEM medium, and co-transfected on the following day with 32 µg of pLenti CMV-GFP-TAV2A-LUC Hygro, 12.5 μg of pMDLg/pRRE (Addgene #12251), 6.25 μg of pRSV-Rev (Addgene #12253), and 9 µg of spike plasmid. Then, 12 h before transfection, the medium was replaced with complete ISCOVE; 30 h after transfection, the supernatant was collected, clarified by filtration through membranes with a 0.45 μm pore size, and concentrated by centrifugation for 2 h at 20,000 rpm using an SW32Ti rotor. Viral pseudoparticle suspensions were aliquoted and stored at −80 °C.

For neutralization assay with pseudotyped particles, HEK293TN-hACE2 cells were transduced with 0.1 MOI of SARS-CoV-2 pseudovirus previously incubated with a serial threefold dilution of serum to obtain a 7-point dose–response curve. Luciferase assay using the Bright-Glo™ Luciferase System (Promega, Madison, WI, USA) and an Infinite F200 plate reader (Tecan, Maennedorf, Switzerland). Measured relative light units (RLUs) were normalized to controls, dose–response curves were generated, and the neutralization dose 50 (ND_50_) was calculated by nonlinear regression curve fitting using GraphPad Prism software.

### 2.7. Necropsy Procedure

On days 3 and 7 after the viral challenge, four animals from each group were euthanized, and their lungs, nasal cavities, and tonsils were submitted for histological evaluation. Sections of the lungs (left cranial, left caudal, right cranial, right middle, right caudal, and right accessory, sampled in a standardized, proximal, and distal orientation from each lobar hilum), tonsils, and nasal cavities were placed in 10% neutral buffered formalin.

### 2.8. Pathological Studies

The nasal cavity was decalcified using an EDTA solution prior to trimming and embedding in paraffin wax. All tissue sections were stained with H&E. The slides were converted to digital images using a Hamamatsu NanoZoomer S360 and viewed with NDP.view2 software. The tissue sections were evaluated independently by two professionally qualified pathologists, who were blinded to the treatments and groups to prevent bias. Changes were scored subjectively as 0 (normal), 1 (minimal), 2 (mild), 3 (moderate), or 4 (marked), using a predefined scoring system.

RNAScope (an in situ hybridization method used on formalin-fixed, paraffin-embedded tissues) was used to identify the presence of SARS-CoV-2 virus histological tissues collected from animals after the cull. Tissues were stained using the Leica Bond RXn (Leica Biosystem, LB) automatic staining machine. Briefly, tissues were pretreated with hydrogen peroxide for 10 min (at room temperature), target retrieval for 15 min (95 °C), and protease III for 15′ (40 °C) (all Advanced Cell Diagnostics, ACD). An LS V-nCoV2019-sense probe (ACD) was incubated on the tissue for 2 h at 42 °C. Amplification of the signal was carried out following the RNAScope protocol (ACD) and the BOND refine RED detection system for chromogenic visualization and hematoxylin counterstaining (LB). The score was assigned as follows: occasional single or group cell staining (1+); scattered positive staining (2+); frequently scattered staining (3+); marked, patchy-to-diffuse staining throughout the tissue (4+).

### 2.9. Statistical Analysis

The Mann–Whitney test was utilized where indicated. All analyses were performed in GraphPad Prism 8.0.2 (Dotmatics, Boston, MA, USA).

## 3. Results

### 3.1. COVID-eVax-Induced Immune Responses in Ferrets

Eight animals per group were administered with COVID-*e*Vax using a prime-only regimen (group 1, 42 days pre-challenge) or a prime–boost regimen (group 2, 42 and 14 days pre-challenge), intramuscularly, with 400 μg of plasmid, which is 1/5 of the maximal human dose (NCT04788459). A control group of eight animals was housed in parallel. Adverse events or effects were not observed following COVID-*e*Vax administration.

Seroconversion was observed in 6/8 ferrets in group 1 and 2/8 ferrets in group 2 on day 14 (Figure 1A).

On day 0, the number of animals with a detectable antibody response increased only in group 2, where six out of eight ferrets showed a relevant antibody titer (>10), while only four of the eight animals in group 1 showed a detectable antibody titer.

Analysis of the effector T-cell response induced by the RBD vaccine was assessed by T-cell ELISpot assay on day 14 and day 0 (Figure 1B). On day 14, IFN-γ-secreting spots were detected in half of the animals in group 2, while none of the unvaccinated ferrets showed a signal (group 3). On day 0, the average number of IFN-γ-secreting cells decreased in group 1, although a signal was detected in two additional animals, suggesting different kinetics of T-cell induction. Group 2, receiving prime–boost vaccinations, showed an increase in the average signal, and 7/8 animals scored positive, while control animals (group 3) lacked immune responses at this time point.

### 3.2. Protection against SARS-CoV-2 and Neutralization of VOCs

Ferrets were infected with the G614D strain, and viral RNA was measured periodically by nasal washing and pharyngeal sampling on days 1, 3 5, and 7 pc (Figure 2).

Comparable amounts of SARS-CoV-2 viral RNA were detected across the groups on days 1 and 3. However, by day 5 there was a decrease in the viral RNA in vaccinated ferrets, and on day 7, group 2 showed a reduction in the amount of viral RNA (*p* = 0.01), with levels of three of the four ferrets being below the limit of quantification. There was therefore evidence for reduced SARS-CoV-2 viral RNA in ferret nasal washes due to prime–boost vaccination. The three animals with undetectable viral shedding were those with the highest ELISpot values (Figure 1B), suggesting a correlation between T-cell response and control of viral replication.

We then investigated the ability of antibody responses induced by COVID-*e*Vax vaccination to neutralize VOCs. To this end, sera collected on day 0, before viral infection, and on day 7 pc were analyzed by pseudovirus neutralization assay against the D614G and two relevant VOCs: Delta (B.1.617.2) and Omicron (B1.1.529) (Figure 3).

On day 0, plasma samples from prime-vaccinated ferrets (group 1) were unable to neutralize any viral strains, although minimal activity against the D614G strain was observed in three ferrets (Table 1).

By contrast, prime–boost vaccination (group 2) resulted in 3/8 neutralizing sera that cross-reacted against Delta, along with 2 against the Omicron variant. Considering the low neutralizing activity, a higher number of responses was observed in five ferrets, although the cross-reactivity against Omicron was still limited (Table 1). Neutralization was not detected in unvaccinated animals at any time point. Although limited to a few observations, it is interesting to note that viral challenge induced a neutralization titer only in vaccinated animals, but not in naïve ferrets (red values in Table 1); by contrast, high neutralizing titers against all strains decreased at day 7—at least in the peripheral blood.

### 3.3. Clinical Signs

Temperature changes throughout the challenge phase of the study were consistent between treatment groups, and displayed the typical diurnal cycle seen in ferrets. No fever (+1 °C above baseline) was detected for any of the animals pc. As with the vaccination phase, daily fluctuations were as a result of increased or decreased activity levels prior to temperature measurements, and represented natural changes to body temperature in a social housing environment. The low temperature recorded for all groups on study day 1 is likely to be a result of temperatures being taken following sedation, and not an effect of SARS-CoV-2 challenge. Following the viral challenge, some animals showed a slight body weight reduction in the vaccination phase; however, ferrets housed in single-sex groups show fluctuations in body weight related to seasonal physiology [14]. For this reason, weight loss in the absence of other clinical signs is not a cause for concern. No clinical signs attributable to the viral challenge were observed following intranasal challenge with 5 × 10^6^ PFU of SARS-CoV-2 isolate Victoria 2020. None of the animals showed signs of fever or weight loss. No evidence of immune enhancement pc was observed in groups 1 and 2.

### 3.4. Histopathology

On days 3 and 7 pc, four animals from each group were euthanized, and their lungs, nasal cavities, and tonsils were submitted for histological evaluation (Figure 4).

In the lungs, the changes observed were primarily low-grade. Mixed inflammatory cells were seen infiltrating the bronchi and bronchioles; these included a variable presence of neutrophils in the walls and lumen, and primarily mononuclear cells accumulating both within and around the walls—often diffusely distributed, and sometimes focal. Perivascular lymphocytic cuffing involving some vessels was noted. Within the alveoli, scattered foci of inflammatory cells were present in low numbers, primarily comprising mononuclear cells—occasionally with neutrophils—filling alveolar spaces and walls.

The severity of these changes in both the lungs and nasal cavities of each animal was scored subjectively, and the results are shown in Table 2.

In the nasal cavity, lesions were noted with increased prominence as compared to the lungs, although the severity of these changes was still considered low-grade. They comprised multifocal degeneration, attenuation, and necrosis of the respiratory epithelial cells lining the cavity; associated with this was an inflammatory cell infiltration of neutrophils and lymphocytes in the propria mucosa, with degenerate epithelial cells and neutrophils admixed with mucus, forming an exudate in the nasal cavity’s lumen.

To correlate immune responses and viral replication in the respiratory tract, viral RNA staining was carried out (Figure 4). In the lungs, viral RNA staining was not observed in any animal from any group at either time point. In the nasal cavity, at day 3 pc, viral RNA was not observed in any animal in the unvaccinated group; one animal in group 1 (one dose) and three animals in group 2 (two doses) were positive (scores 1, 2, and 3). On day 7 pc, viral RNA was detected in two control animals (scores 2 and 3) and two animals from group 1 (score 1).

## 4. Discussion

In this study, we explored COVID-*e*Vax in ferrets, and characterized its safety and immunological effects. No adverse reactions and no differences in body weight or temperature were observed in the groups during the vaccination phase of the study. The overall results suggest that the vaccine was well tolerated with both the prime and prime–boost regimens. The animals did not exhibit adverse clinical signs or prominent changes in body temperature. The prime–boost vaccination regimen induced seroconversion in 75% of animals, with the highest antibody titers showing neutralization activity (Figure 1A and Figure 3). Importantly, the vaccine also induced a potent T-cell immune response (Figure 1B), thus confirming the data generated in rodents [7].

With the advent of Omicron-derived VOCs endowed with high infectivity in the upper respiratory tract, along with the waning of neutralizing antibody titers induced by approved SARS-CoV-2 vaccines, the control of COVID-19 may indeed rely on the T-cell response. In the event that the neutralizing antibody titer fails to control viral infection, it becomes relevant to keep viral replication under control. The T-cell immune response induced by COVID-*e*Vax in the ferrets resulted in a high frequency of IFN-γ-producing cells, which was correlated with reduced viral RNA (Figure 2). Here, we show a clear correlation between vaccine-induced immune responses and RNA shedding in a time-course study of SARS-CoV-2 infection. Our data suggest that the ferret model system recapitulates the natural course of a mild viral infection that is further ameliorated by the DNA-EP vaccine. The reduced viral shedding was further supported by the lack of histological changes or evidence of viral RNA found by in situ hybridization (Figure 4 and Table 2). The ferret model provides a straightforward way to measure the efficacy of vaccines through the reduction in viral RNA, while sera can also be used to screen for efficacy against VOCs. As observed with mRNA vaccines [15], the neutralizing antibodies induced by COVID-*e*Vax were less efficient against the Delta and Omicron variants. Interestingly, the viral challenge in animals where the vaccine induced a low neutralizing titer can elicit a neutralizing antibody titer against the VOCs (Table 1), while the presence of high titer is not increased by the viral challenge. Importantly, we observed a vaccine-induced reduction in viral shedding.

Despite these promising results, this study has several limitations. The vaccine was administered with one or two doses, following the criteria adopted for approved mRNA or Ad vaccines. However, it is interesting to note that the ZyCoV-D vaccine was approved with a three-dose regimen, suggesting that an additional dose is required for the DNA vaccine platform to be completely efficacious. Indeed, in a rat model, we observed a steep increase in the neutralizing antibody titers when a third vaccine dose was administered [7]. The elicited antibodies were equally able to neutralize multiple variant pseudotyped viruses. An additional difference with the approved DNA vaccine is the use of the RBD domain in COVID-*e*Vax and the full-length spike protein in ZyCoV-D. As we reported in mice, COVID-eVax, which targets only the RBD region, induces a strong CD8 response but a limited CD4 response [7]. This is not an intrinsic property of DNA-EP delivery systems, as we have shown relevant CD4 T-cell responses using the DNA-EP technology to deliver a different vaccine [11]. The likely presence of additional CD4 epitopes in the full-length spike protein of approved vaccines may contribute to the quality of antibodies, in terms of antibody titers and viral neutralization.

Notably, COVID-*e*Vax was recently evaluated in a phase I clinical trial (NCT04788459) of 68 enrolled subjects showing a potent T-cell immune response but low neutralizing antibody titers (manuscript submitted). The similarity of the obtained results suggests that the ferret is indeed a good predictive animal model for evaluating immunotherapies against respiratory viruses.

In conclusion, this study shows a safe profile of COVID-*e*Vax and a robust immunogenicity response in ferrets. The T-cell-skewed immunological response observed may be crucial for cross-reactivity with VOCs to protect against severe disease.

## Figures and Tables

**Figure 1 vaccines-10-01178-f001:**
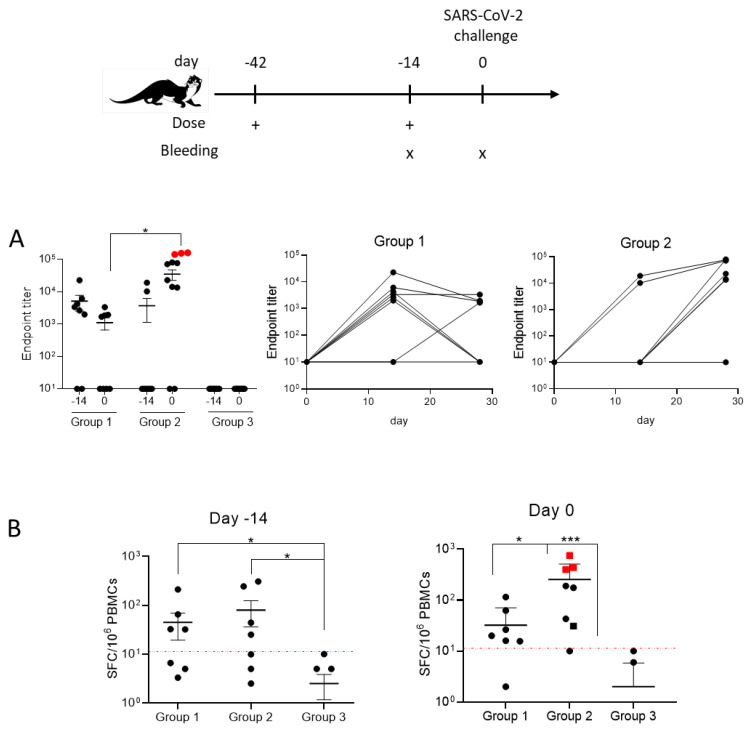
Immune responses induced by COVID-*e*Vax. Groups of ferrets (*n* = 8) were vaccinated with one dose on day 42 (one dose, group 1), with a second dose on day 14 (two doses, group 2), or left untreated (Group 3). Sera and PBMCs were collected on day 14 and day 0 before DNA vaccination. (**A**) Endpoint titer values of binding antibody measured by ELISA at day 0, and antibody titers over time. (**B**) T-cell responses measured by ELISpot assay. PBMCs were stimulated with the RBD peptide pool as described in the Materials and Methods. IFN-γ-producing spot-forming colonies (SFCs) were counted, and DMSO values were subtracted; animals in group 3 with values below 1 are not reported in the logarithmic scale. Values in red are from the animals in group 2 with viral RNA below the LLOQ in Figure 2. Mann–Whitney tests were conducted: * *p* < 0.05, *** *p* < 0.005.

**Figure 2 vaccines-10-01178-f002:**
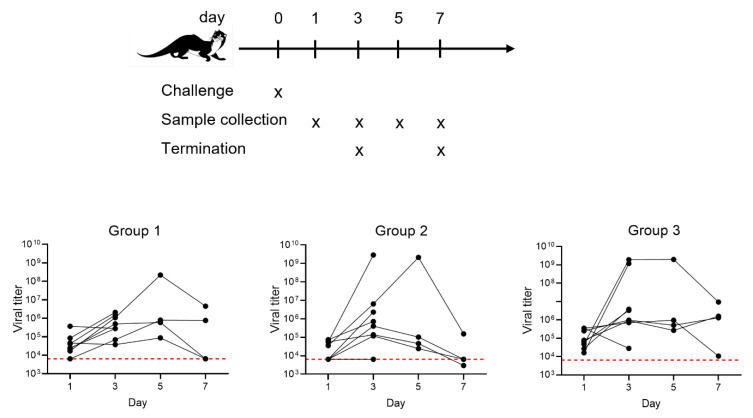
Detection of SARS-CoV-2 RNA in infected ferrets. Viral RNA in unvaccinated or COVID-*e*Vax-vaccinated ferrets was quantified by RT-PCR in nasal washes over time. The lines plotted are the geometric mean genome copies/mL. The red line indicates the lower limit of quantification (LLOQ).

**Figure 3 vaccines-10-01178-f003:**
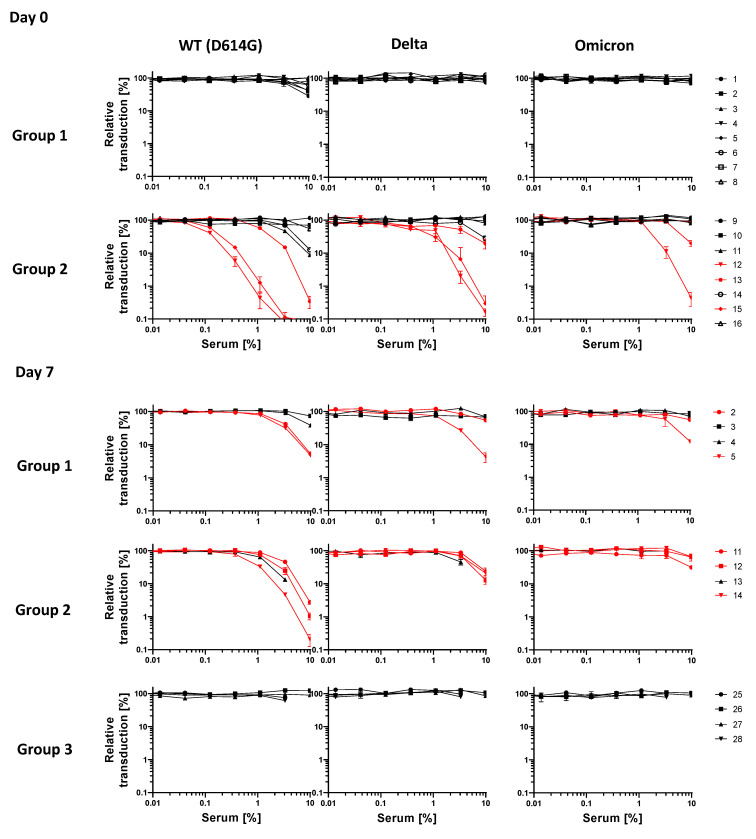
Dose–response curve representing the neutralization activity of plasma against SARS-CoV-2 pseudovirus carrying the spike protein of WT (D614G) virus or variants (Delta and Omicron). The sera analyzed were collected on day 0 (challenge) and day 7 (termination) from vaccinated and unvaccinated ferrets.

**Figure 4 vaccines-10-01178-f004:**
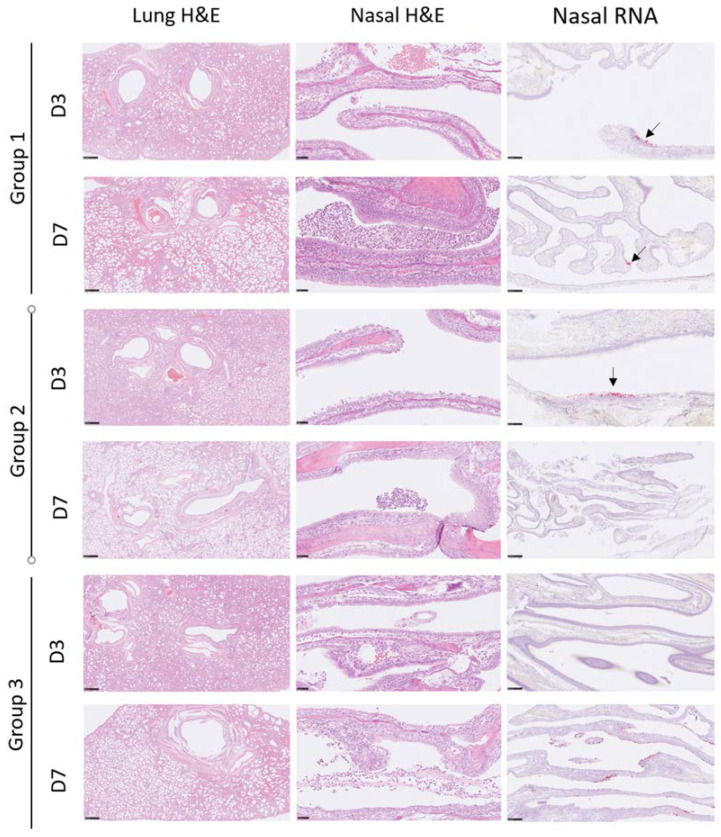
Histopathological changes in the lung and nasal cavity, and viral RNA staining in the nasal cavity. Column one: lung H&E, with low-grade inflammatory infiltrations. Column two: nasal H&E, with inflammatory exudation in the lumen and respiratory epithelial degeneration. Column three: nasal RNA, with staining of SARS-CoV-2 RNA in the epithelia (arrows highlight smaller patches of viral-positive cells).

**Table 1 vaccines-10-01178-t001:** Neutralization titers against VOCs. Not neutralizing sera (NN).

			Day Zero			Day Seven	
	Fer.	Wt	Delta	Omi	Wt	Delta	Omi
**1 dose**	1	**51**	NN	NN	-	-	-
	2	**35**	NN	NN	**109**	NN	NN
	3	NN	NN	NN	NN	NN	NN
	4	NN	NN	NN	** 36 **	NN	NN
	5	NN	NN	NN	** 139 **	** 166 **	** 77 **
	6	NN	NN	NN	-	-	-
	7	**35**	NN	NN	-	-	-
	8	NN	NN	NN	-	-	-
**2 doses**	9	NN	NN	NN	-	-	-
	10	NN	NN	NN	-	-	-
	11	**93**	NN	NN	**100**	** 45 **	** 40 **
	12	**3062**	**507**	**153**	**154**	**64**	NN
	13	**224**	**44**	NN	**211**	**101**	NN
	14	**64**	**46**	NN	**407**	**60**	NN
	15	**1974**	**607**	**46**	-	-	-
	16	NN	NN	NN	-	-	-
**not vac.**	17	NN	NN	NN	-	-	-
	18	NN	NN	NN	-	-	-
	19	NN	NN	NN	-	-	-
	20	NN	NN	NN	-	-	-
	21	-	-	-	NN	NN	NN
	22	-	-	-	NN	NN	NN
	23	-	-	-	NN	NN	NN
	24	-	-	-	NN	NN	NN

**Table 2 vaccines-10-01178-t002:** Severity scores for lesions and the presence of viral staining in the lungs and nasal cavity.

Group	Days Pc	Lungs						Nasal Cavity		
		Bronchial Inflammation (Bronchitis)	Bronchiolar Inflammation (Bronchiolitis)	PVC	Inflammation of Alveoli	Average Lung Scores	Viral Staining Intensity	Epithelial Inflammation/Necrosis	Lumenal Exudate	Viral Staining Intensity (SARS-CoV-2)
							(SARS-CoV-2)			
Group 1		1	2	0.5	1	1.1	0	2	1	1
(one dose)		0.5	1	1	1	0.9	0	2	0	0
		0.5	1.5	1.5	1	1.1	0	1	0	0
	3	1.5	2	1.5	1.5	1.6	0	0	0	0
		1.5	1.5	1.5	1.5	1.5	0	1	0	0
		0.5	2	1.5	1	1.3	0	2	2	0
		1.5	1.5	2	1	1.5	0	2	2	1
	7	1.5	0.5	0.5	0.5	0.8	0	0	0	1
Group 2		1.5	2.5	1.5	2	1.9	0	0	0	2
(two doses)		1.5	2	0.5	1	1.3	0	2	1	3
		1	1.5	1	1.5	1.3	0	1	1	1
	3	1.5	2	1.5	2	1.8	0	0	0	0
		1.5	1	0.5	1	1.0	0	1	1	0
		1.5	1.5	0.5	1	1.1	0	1	1	0
		1	1	0.5	1	0.9	0	1	1	0
	7	0.5	1.5	0.5	1	0.9	0	1	1	0
Group 3		1.5	1.5	0.5	1	1.1	0	0	1	0
(unvaccinated)		1.5	2	1	2	1.6	0	1	1	0
		1.5	2.5	2	1.5	1.9	0	0	0	0
	3	1.5	1.5	1.5	1.5	1.5	0	1	0	0
		1	2	0.5	1	1.1	0	2	2	0
		1	1.5	1	1.5	1.3	0	1	1	0
		1	1	0.5	1	0.9	0	1	1	2
	7	1	1.5	1	1	1.1	0	2	1	3

## Data Availability

Not applicable.

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
