# Peer review of "DNA-Vaccine-Induced Immune Response Correlates with Lower Viral SARS-CoV-2 Titers in a Ferret Model"

_vaccines, 2022, doi:10.3390/vaccines10081178_

Round 1

Reviewer 1 Report

 Compagnone et al. demonstrated the efficacy of the DNA vaccine against SARS-CoV-2 expressing the RBD domain of the SARS-CoV-2 spike protein in a ferret model. Although the technical limitation in the delivery (electroporation) and relatively low efficacy (75% seroconversion rate presumably due to the unoptimized vaccination schedule) of this COVID-eVax, the current work is solid in the robust analysis of the immune response and protection and significantly contributes to the field of the vaccine development for COVID-19 exploring the possible adjuvant effect of EP without limiting immune responses using naked DNA. A correlation between vaccine-induced T cell responses and reduced viral RNA in nasal washing is an interesting observation indicating a relevant immune correlate.

The reviewer has minor technical comments only.

On page 5, line 219;  2.9. Statistical analysis, “Mann–Whitney was utilized where indicated” is described. However, there are more than 2 experimental groups (three groups) over several time points in Figure 2. Please state how the authors corrected the multiplicity of the statistical tests.

On page 2, line 62, …COVID-eVax resulted in a limited neutralization capacity, “It” primes…

On page 12, line 352, … keep viral replication under control, a period instead of a comma here?

Author Response

Dear reviewer, thank you for your comment "state how the authors corrected the multiplicity of the statistical tests". Actually, there are no multiple comparisons. Statistical analysis was conducted comparing one group with another group.

On page 2, line 62 and on page 12, line 352, text was modified accordingly

Reviewer 2 Report

Compagnone and colleagues report the result of the evaluation of the tolerability and the immunogenicity of a DNA vaccine against SARS-CoV2 in a ferret model of infection. The animals were immunized with either a single, or a prime-boost administration of a naked DNA coding for the Spike protein receptor-binding domain (RBD). The DNA was delivered intramuscularly via electroporation; no adverse effects were observed upon either immunization scheme. The immunization in both groups resulted in partial seroconversion, and partial T-cell response as measured by an ELISA against a recombinant RBD and Elispot for IFN-gamma secretion by peripheral blood monocytes stimulated with Spike protein peptides, respectively. Upon SARS-CoV2 challenge, the animals vaccinated according to the prime-boost protocol, showed partial protection against virus shedding, although in all vaccination groups the majority of the animals shed the virus similar to the animals from the control unvaccinated group. The assessment of the neutralization ab profile showed weak cross-reactivity of the antibodies in some animals against the variants of concern, and the breadth of cross-reactivity correlated with the strength of the T-cell response.

Overall, the paper is well written, and presents important data on the characterization of an alternative vaccine platform against SARS-Cov2. The authors need to clarify the following places in the manuscript:

Line 119 states that secondary anti-murine IgG abs were used in ELISA with ferret sera.  If it is not a mistake, provide the rationale.

Lines 229-230. Remove the sentence “To verify…… two groups of eight ferrets were administered with one dose of the vaccine on day -42”. This is confusing, in the previous paragraph you just described the administration schemes of both groups.

Fig.1 B why the signal from the control group (3) is shown only for 3 and 2 animals for the days -14 and 0, respectively, not for all 8 as in the other groups?

Author Response

We thank the reviewer for the useful comments, below point by point answers:

Lane 119: yes, it was a mistake the secondary antibody was an anti-ferret. The text was modified accordingly. 

Lines 229-230 text removed

Fig.1 B thank you for noticing the missing information, to address this point we added in the figure legend the notice "animals in group 3 with values below 1 are not reported in the logarithmic scale".